# Research Progress and Challenges in Laser-Controlled Cleaning of Aluminum Alloy Surfaces

**DOI:** 10.3390/ma15165469

**Published:** 2022-08-09

**Authors:** Jian Deng, Guanrong Zhao, Jieheng Lei, Lin Zhong, Zeyong Lei

**Affiliations:** 1School of Nuclear Science and Technology, University of South China, Hengyang 421001, China; 2School of Mechanical Engineering, University of South China, Hengyang 421001, China; 3School of Electrical Engineering, University of South China, Hengyang 421001, China

**Keywords:** laser, controlled cleaning, aluminum alloy, oxide film, paint layer

## Abstract

Aluminum alloys have been widely utilized in automobiles, aircraft, building structures, and high-speed railways industries due to their excellent structural and mechanical properties. Surface oxide film removal prior to aluminum alloy welding and old paint removal prior to repainting aluminum alloy surfaces are critical factors in ensuring the welding quality and service life of aluminum alloy products. Because of its unique advantages, such as environmental protection and precision control, laser-controlled cleaning has great application potential as a surface cleaning technology in removing oxide films and paint layers on aluminum alloy surfaces. In this paper, the mechanism of laser cleaning of oxide films and paint layers on aluminum alloy is discussed. Furthermore, the impact of various processing parameters such as laser beam power, energy density, scanning speed, and so on is analyzed in detail. After laser cleaning, the corrosion resistance, welding performance, adhesive performance, and other properties of the aluminum alloy are optimized. This paper also discusses several real-time detection technologies for laser cleaning. A summary and the development trend are provided at the end of the paper.

## 1. Introduction

Aluminum alloys are widely used in aerospace, construction, automobiles, high-speed rail, and other industries owing to their low density, good processing performance, high specific strength, and excellent corrosion resistance [1,2,3,4]. However, if there is an oxide film on the surface of aluminum alloy, it is easy to absorb water and oil, so that a large numerous of pores appear in the weld seam during welding, which will seriously affect the formation of the aluminum alloy weld seam and the performance of the welding parts [5,6]. To improve the welding quality, we should strictly remove the oxide films before welding aluminum alloys [7,8]. Furthermore, the surface paint layers on equipment made of aluminum alloy would eventually age, break, and partially peel off after prolonged usage, losing its protective function on the aluminum alloy surface [9]. Therefore, the surface of equipment made of aluminum alloy must be repainted often. Before spraying, a paint-stripping procedure must be carried out, and the substrate should not be harmed [10,11]. For instance, an aircraft needs to be overhauled regularly (typically every 5 to 6 years). During the overhaul, the paint layer in the vicinity of the skin should be removed in order to respray the paint layer on the skin surface and identify fractures and faults in the aluminum alloy skin and structural elements [12,13].

Traditional methods of removing the oxide films or paint layers from the aluminum alloy surfaces mainly include the mechanical and chemical removals. The mechanical removal process, which is more efficient than manual labor, uses scraping, brushing, and shot peening to remove the oxide film or paint layer. However, the mechanical removal is ineffective in removing the oxide films or paint layers on the curved surfaces or intricate structures. Furthermore, the mechanical removal has a detrimental impact on the surface roughness, microhardness, residual stress, and fatigue resistance of the aluminum alloy. In addition, the mechanical removal has some drawbacks, including intense working conditions, extreme noise, and dust pollution [14,15]. Meanwhile, chemical removal uses chemical reagents to remove the oxide films or paint layers, showing positive results. However, the process is time-consuming, difficult to perform, the substrate is fragile, the cleaning waste liquid is difficult to control, and the environment is highly contaminated [16,17]. In contrast to the conventional decontamination process, the laser-controlled cleaning method cleans the material surface by acting on it with concentrated laser light, which quickly vaporizes or removes the material off the surface. It may be used on various materials and enables nonabrasive, noncontact, and nonthermal surface cleanings [18,19]. It has some benefits, including high control precision, high decontamination efficiency, good decontamination quality, broad application potential, minor substrate damage, an environment-friendly process, easy automation, and cheap operating costs [20,21]. It is considered to be the most reliable and efficient green decontamination technology [22].

A. L. Schawlow invented laser cleaning in 1965 by using a ruby-pulsed laser to remove black pigments from paper in a targeted manner [23]. Since then, the laser-controlled cleaning technology has been applied to a variety of cleaning tasks, including the cleaning of artworks [24], artifacts [25], and glass surfaces [26], the removal of radionuclides [27], the decarburization of tire-abrasive surfaces [28], the removal of rust from carbon steel or stainless steel surfaces [29,30], and the removal of oxide films and paint layers from aluminum alloy surfaces [31,32].

Due to its excellent application market and prospects, the laser-controlled cleaning of aluminum alloy surfaces has recently attracted study attention [33,34,35]. Understanding the laser cleaning mechanisms, mastering the selection rules and control techniques of laser process parameters, ensuring that the surface oxide film or paint layer is removed without causing damage, and optimizing the substrate performance are all necessary for performing laser-controlled cleaning of aluminum alloy surfaces. Thus, this paper reviews the research on laser-controlled cleaning of aluminum alloy surfaces, with a focus on the laser cleaning mechanisms, process parameter optimization, substrate performance optimization after cleaning, real-time detection of laser cleaning, and prospects for future research directions.

## 2. The Mechanism of Laser Cleaning

The bonding force between the oxide film or paint layer on aluminum alloy surfaces and the substrate primarily includes the covalent bond, double dipole interaction, capillary action, hydrogen bond, van der Waals force, and electrostatic force [36,37]. Laser cleaning technology exploits the advantage of a laser beam with a large energy density and spatial coherence to directly vaporize the oxide film or paint layer on the aluminum alloy surface or to remove the adhesive force holding it to the substrate [38]. The action mechanism mainly includes the ablation gasification, vibration, and explosion stripping effects [39,40] (Figure 1). Laser cleaning is a highly complex physical and chemical process, and several coupling effects are typically involved in cleaning each substance [34].

### 2.1. Ablation Gasification Effect

When a high-energy laser beam strikes an oxide film or paint layer on the aluminum alloy surface, some light is reflected off the surface. The remaining light is absorbed by the oxide film or paint layer, which also collects the surface energy and quickly raises the temperature. When the energy received by the oxide film or paint layer exceeds its gasification point, the oxide film or paint layer on the aluminum alloy surface instantly vaporizes and volatilizes, detaching from the aluminum alloy surface and realizing cleaning [41,42]. In parallel, during the gasification of the oxide film or paint layer, the gasification gas and the surrounding air are ionized to form plasma, which then gathers into a plasma group. While the plasma group reaches a certain size, it produces a high-pressure shock wave on the material surface. Under the shockwave, the oxide film or paint layer are detached from the aluminum alloy surface [43].

### 2.2. Vibration Stripping Effect

When the laser acts on the oxide film or paint layer on the aluminum alloy surface, light energy is converted into heat energy, and the oxide film or paint layer expands rapidly [44,45]. The thermal stress produced during the expansion process is stronger than the chemical bond or van der Waals force between the materials and exceeds the adhesion force between the oxide film or paint layer and the substrate. Consequently, the oxide film or paint layer can be peeled off from the surface of the substrate [46,47]. In addition, when the laser beam travels through the oxide film or paint layer and eventually reaches the substrate surface, a portion of it is reflected. The incident laser beam causes interference inside the oxide film or paint layer during reflection. As a result, high-energy resonant waves are created at the interference, whose energy impacts the oxide film or paint layer and accelerates the removal of this layer [13].

### 2.3. Explosion Stripping Effect

The pores and gaps of the oxide film or paint layer are typically enriched with air or water. The air or water in the pores and gaps in the aluminum alloy surface absorbs energy when exposed to a high-intensity laser, rapidly heating up, expanding, or exploding [48]. When the pressure caused by the explosion expansion or impact force overcomes the adhesion force between the oxide film or paint layer and the substrate, the oxide film or paint layer breaks and may be peeled off the aluminum alloy surface [49].

## 3. Optimization of Laser Process Parameters

The effect of the laser-controlled cleaning is mainly evaluated according to the removal rate of oxide films and paint layers from the aluminum alloy surface and the damage degree of the substrate. Optimizing the laser process parameters is crucial to realize the laser-controlled cleaning of the aluminum alloy surface and guarantee the effect of the removal. The laser process parameters are primarily divided into laser beam parameters (including beam power, energy density, repetition frequency, and pulse width) and laser space parameters (including cleaning speed, scanning speed, and defocus distance).

### 3.1. Laser Beam Parameters

The laser beam parameters refer to the property parameters of the laser beam, including the laser beam power, energy density, repetition frequency, and the pulse width. Table 1 shows the research on the optimization of the laser beam parameters. As shown in Table 1, the nanosecond pulsed fiber laser and the Nd: YAG laser are generally used to remove the oxide films and paint layers on the surface of an aluminum alloy. The change of the laser beam parameters significantly influences on the effect of the laser cleaning. For any specific coating, there is an ideal cleaning parameter value, which is determined by oxygen removal rate, temperature fluctuations, surface roughness changes, etc.

#### 3.1.1. Laser Beam Power

One of the crucial factors affecting the effect of laser cleaning is the laser beam power, which determines the energy of the aluminum alloy surface irradiated by the laser in one unit area and one unit time. The existing studies have shown that the removal effect of oxide films and paint layers from the aluminum alloy surface is gradually improved with increasing laser beam power. However, when the power rises to a certain extent, the oxide film and paint layer are removed, and the aluminum alloy substrate is ablated [12,50,51,52,53,54,62]. Shao et al. [50] used a pulsed laser to clean the oxide film from a 7075 aluminum alloy surface, and the laser beam power was increased from 16 to 160 W. The experimental results displayed that the oxide film on the aluminum alloy surface was removed entirely when the laser beam power was 80 W. The oxide film only cracked when the power was lower than 80 W since there was not enough power. The microcracks started to emerge on the surface when the laser beam strength topped 80 W. As the power rose, the fissures became deeper and deeper, seriously damaging the substrate (Figure 2).

Similarly, Zhu et al. [51] and Tong et al. [52] discovered that the oxide film on the aluminum alloy surface could be successfully removed by raising the laser beam power. The aluminum alloy is melted excessively with continuedly increasing power, and the oxide film is observed to absorb some of the laser energy. To be effectively removed based on the analysis of the oxygen element contents, the substrate absorbs the remaining energy, which leads to the secondary oxidation. During the cleaning process, the high-purity argon can be used as a protective gas to isolate the oxygen in the air and prevent secondary oxidation on the aluminum alloy surface [62]. The anodic oxide film on the aluminum alloy surface can be successfully removed by altering the laser beam power. The removal rate is positively correlated with the laser beam power [53]. Zhao et al. [54] used a pulsed fiber laser to clean a polyacrylate resin-based paint layer with a thickness of approximately 50 μm on a 2024 aluminum alloy surface. The laser beam power was raised from 10.5 to 25.5 W. The three-dimensional morphology and roughness analysis of the aluminum alloy surface after cleaning indicated that the coating layer on the aluminum alloy surface was gradually removed by raising the laser beam power. When the power reached 16.5 W, the coating layer was removed, but a slight melting was visible on the substrate surface. Subsequently, Qiu et al. [12] cleaned the paint coating on the 2024 aluminum alloy surface by using a pulsed laser and successfully obtained a laser paint removal threshold of 138.19 W and a damage threshold of 556.17 W for the substrate.

#### 3.1.2. Laser Energy Density

The laser energy density refers to the energy of the laser output pulse divided by the area of the laser output spot, and is determined by the average power, repetition frequency, and laser spot area. The change in the laser energy density significantly impacted the removal effect of the oxide films and paint layers from aluminum alloy surfaces. Many studies have shown that the removal effect of oxide film on the aluminum alloy surface initially rises and then falls with increased energy density [13,33,55,56,57,58,59]. The energy-dispersive X-ray spectroscopy and the X-ray photoelectron spectroscopy (XPS) were used to detect the surface composition of the aluminum alloy. This technology revealed that the initial oxide film on the aluminum alloy surface could be successfully removed with increased energy density. However, due to the thermal oxidation, the amount of freshly produced Al_2_O_3_ rises [33,55,56]. As shown in Figure 3, the initial sample surface was Al_2_O_3_ and Al matrix. The composition of the surface does not change when the laser energy density increases after laser cleaning. However, the relative concentration of Al_2_O_3_ initially fell and then climbed, along with the shift in oxygen content, shown in Figure 4. When the laser energy density is 17.5 J/cm^2^, the Al_2_O_3_ composition reaches the lowest level. It proves that the laser cleaning can remove the native Al_2_O_3_. Above 17.5 J/cm^2^, the Al_2_O_3_ concentration rose. Zhu [13], Miao [57], and Zhang [58] confirmed the impact of the laser paint removal by altering the laser energy density. Their results showed that the paint layer may be completely removed without damaging the aluminum alloy surface by selecting the appropriate energy density parameters [59]. When the energy density value is below the optimal cleaning value, the removal effect is gradually improved with higher energy density. However, when the energy density exceeds the value of the optimal cleaning, the surface of the aluminum alloy substrate is ablated after the removal of the paint layer [60].

#### 3.1.3. Laser Repetition Frequency

The repetition frequency refers to the number of laser pulse outputs in per unit time, or the number of pulses repeated within one second. The pulse frequency directly determines the speed of the laser cleaning. Yang et al. [61] employed a pulsed laser to remove an oxide film from a 6106–T6 aluminum alloy surface and raised the repetition frequency from 40 to 140 kHz. The analysis of the morphology, oxygen content, and distribution properties of the cleaned aluminum alloy surface showed that the surface morphology gradually transitioned from independent pits with a certain spacing to a tightly packed honeycomb form with increased pulse repetition frequency. Furthermore, a lower repetition frequency results in larger residual oxygen levels. However, when the repetition frequency exceeds 80 kHz, the amount of the residual oxygen is significantly reduced. In terms of morphology, when the repetition frequency is low, the impact craters formed by laser cleaning are scattered. The impact craters pack tightly, and the cleaning area expands when the repetition frequency rises to a certain value. At this time, the oxide film on the aluminum alloy surface has nearly wholly peeled off. Similarly, by analyzing the three-dimensional topography and roughness of the aluminum alloy surface after being cleaned, it was indicated that the cleaning effect of the aluminum alloy surface initially rose and then declined with an increase in pulse frequency. In contrast, the surface roughness initially rose and then reduced [54].

#### 3.1.4. Laser Pulse Width

The laser pulse width refers to the duration of maintaining the laser at a specific power. The research results showed that the laser cleaning effect of the oxide film on an aluminum alloy surface initially increased and then dropped as the pulse width is reduced. When the pulse width is large, the removal of the oxide film depends on the ablation gasification process. However, when the energy generated by the laser beam power does not reach the gasification point of the oxide film, the cleaning effect is poor. When the pulse width is reduced to an appropriate parameter value, an optimal cleaning effect is realized because the laser cleaning depends on the coordinated action of ablation gasification, explosive stripping, and vibration stripping, which can effectively remove the oxide film without damaging the substrate. However, the pits and secondary oxidation are produced as a result of the laser damage to the substrate when the pulse width is reduced [53].

### 3.2. Laser Space Parameters

The space parameters of a laser refer to the relative displacement between the laser cleaning head and aluminum alloy surface, as well as the speed at which the laser beam moves over the surface, including the laser cleaning speed, the laser scanning speed, and the laser defocus distance. Table 2 summarizes the research on the optimization of laser beam parameters. As shown in Table 2, the change of the laser space parameters tremendously influences the laser cleaning effect. As a result, altering the laser process parameters can result in cleaning that is controlled by lasers. It is noted that there is a distinct ideal cleaning parameter value for the different aluminum alloy compositions and surface coatings.

#### 3.2.1. Laser Cleaning Speed

The laser cleaning speed refers to the moving speed of the laser cleaning head during the cleaning process. The change of the cleaning speed primarily affects the laser irradiation time of the oxide film and paint layer per unit area. Studies have shown that the removal effect of oxide film from an aluminum alloy surface initially increases and then drops with the decrease of the laser cleaning speed. The oxide film or paint layer can be removed from the aluminum alloy surface with minimal surface roughness by using an appropriate laser cleaning speed. Simultaneously, with a gradual increase of the cleaning speed, the roughness value of the aluminum alloy surface increases, making the aluminum alloy surface rougher than the original sample [51]. However, when the cleaning speed is slowed down, the aluminum alloy substrate is damaged by laser thermal stress [63]. As shown in Figure 5, when the cleaning speed was 10.4 mm/s, the aluminum alloy surface seemed rougher, and yet displayed fewer annular structures and holes. When the cleaning speed was dropped to 4.1 mm/s, the surface of the aluminum alloy was basically flat with a few holes. As the cleaning speed continued to drop to 1.0 mm/s, the number of holes clearly increased. As a result, there is a cleaning and damage threshold for the laser cleaning of the oxide film on an aluminum alloy surface. If the relevant parameters of the laser exceed the damage threshold, the substrate will be damaged even though the oxide film is removed.

#### 3.2.2. Laser Scanning Speed

The change in the laser scanning speed is mainly reflected by a shift in the lapping rate of a single laser spot. With increasing scanning speed, the number of laps is reduced, and less paint and oxide film are removed from the aluminum alloy surface. When a pulsed laser with a low scanning speed is used to remove the oxide film on the surface, the spot overlap rate becomes high. After the oxide film is removed, the substrate surface is excessively cleaned, leading to significant ablation and thermal effects that permanently harm the aluminum alloy substrate. In other words, the cleaning effect of the oxide film on the surface of aluminum alloy initially increases and then drops with an increase in the scanning speed. As shown in Figure 6, when the laser scanning speed is 20 mm/s, the surface resembles plowed soil. When the laser scanning speed is 200 mm/s, there are still a lot of oxides on the surface. As the scanning speed rises to 1000 mm/s, the micro-cracks on the aluminum alloy surface vanish, the dense structure is formed, and the oxidized layer is removed. When the laser scanning speed reaches 3000 mm/s, the fish scale phenomena begin to appear on the surface [50]. Therefore, when the scanning speed is suitable, the oxide film will be removed without damaging the substrate. When the scanning speed increases, the splicing between two adjacent pulses is incomplete, the molten pool formed by the previous focus cools prematurely, the fish scales appear on the surface, and the oxide film remains partially intact [53]. Zhang and Miao et al. concluded that using a pulsed laser to remove an acrylic resin paint layer on a 7075 aluminum alloy surface could be successful [54,57].

#### 3.2.3. Laser Defocus Distance

The defocus distance is the distance between the laser focus and the action alloy surface. The laser defocus distance could be altered by adjusting the distance between the laser head and the aluminum alloy surface. The removal effect of the oxide film and paint layer from the aluminum alloy surface is directly related to the laser defocus distance. The spot size and power density will alter depending on defocus distance at which the laser interacts with the oxide film and paint layer. Zhu et al. [64] studied the impact of defocus distance on the paint layer removal mechanism by using the defocus distance in both directions of inside and outside the Rayleigh length (0.61 mm). When the defocus distance is 0 mm, the paint layer is primarily removed under thermal stress, melting, and gas impact. At this time, it is suitable for removing the surface paint since the power density is at its peak. When the defocus distance varied from −4 mm and +4 mm, the removal effect and mechanisms are similar to each other, and thermal stress and melting evaporation primarily accomplish the removal. The procedure of paint removal is controllable and suitable for removing the paint layer near the interface. Similarly, Zhu et al. [65] studied how defocus distance affected the removal of paint layers by employing the defocus distance in both directions inside and outside the Rayleigh length (1.84 mm). They observed the surface after the laser cleaning and found that when the absolute value of the defocus distance is the same, the cleaning effect is the same. When the defocus distance is 0 mm, the energy density is the maximum, and the amount of heat absorbed by the paint layer is the greatest, yielding to the best cleaning results. The above findings confirm that the ideal defocus distance is within Rayleigh length.

We have gained a thorough understanding of the essence of laser cleaning by summarizing the influence of laser beam and space parameters on the cleaning effect: a laser irradiated on the object surface, and a series of physical or chemical reactions occur to remove contaminants. The contaminant removal is affected by the energy and duration of laser irradiation. The laser process parameters, in particular, determine the irradiation energy and time. For instance, the laser power, repetition frequency, and defocus amount all work together to determine the laser energy (energy density) received by the unit area contaminant. The scanning speed, cleaning speed, and pulse width all work together to determine the action duration between the laser and contaminants in the same area region. These factors are linked to the laser cleaning process, and their impacts on the cleaning effect are connected. When the contaminants are removed under the gasification mechanism, on the one hand, the energy density should be appropriately reduced to avoid excessive irradiation energy. On the other hand, the scanning and cleaning speeds should be appropriately increased, and the pulse width should be decreased to prevent exposing contaminants to laser irradiation for an extended period and to avoid over-ablating the substrate. When the contaminants are removed under the peeling mechanism, on the one hand, the energy density should be appropriately raised to improve the peeling efficiency. On the other hand, the scanning and cleaning speed should be slowed down, and the pulse width should be increased to prolong the irradiation duration of pollutants in the same location and improve the cleaning efficiency to ensure substrate safety.

The formula of laser energy density *ε* is as follows:(1)$$\varepsilon =\frac{4P}{\pi D^{2}f}$$
where *ε* is laser energy density, *P* is laser power, *D* is laser spot diameter, and *f* is repetition frequency.

The laser energy density is a crucial laser coupling parameter. According to the above literature, the recommended energy density range for using a laser to remove an oxide film from the surface of an aluminum alloy is 7.1–17.5 J/cm^2^, and the recommended energy density range for using a laser to remove paint layers from the surface of an aluminum alloy is 3.2–25 J/cm^2^.

## 4. Optimization of Substrate Performance

When utilizing a laser to remove the oxide film or paint layer from an aluminum alloy surface, a controlled cleaning can be achieved by adjusting the laser process parameters, such as laser beam power, energy density, and scanning speed. Interestingly, optimizing the laser process parameters can efficiently remove oxide films or paint layers from aluminum alloy surfaces and improve the performance of the substrate.

### 4.1. Corrosion Resistance

The corrosion resistance of the aluminum alloy surface is improved after the oxide film or paint layer is removed by using a laser, and the corrosion resistance of the aluminum alloy is better than that obtained via mechanical cleaning. By comparing the substrate corrosion resistance and the mechanisms after the mechanical and laser cleaning, the research found that the aluminum alloy corrosion resistance after laser cleaning is higher than that obtained after mechanical cleaning because the grain size did not change [66]. However, the aluminum alloy was thermally oxidized after the laser cleaning, the surface was remelted, and the initial MgAl_2_O_4_ on the aluminum alloy surface was decomposed to obtain a new nanostructured layer comprising MgO and Al_2_O_3_ [67], with making the element distribution uniform and the refining of the grains [68]. This result can be clearly seen from the statistics chart of the grain size of aluminum alloy substrate after the laser cleaning (Figure 7), the average grain size of the whole region was 7.98 μm, and the percentage of grains with less than 10 μm accounted for 65% of the total. The average grain size of the upper region was 7.30 μm, and 73% of the total number of grains were smaller than 10 μm in diameter. These results lead to the difference between the grain boundary and intragranular composition minimal, lowering the corrosion current density [69]. Zhu et al. [13] conducted electrochemical experiments and verified that the laser cleaning does not reduce the corrosion resistance of aluminum alloy surfaces, and that the improvement of corrosion performance is related to energy density.

### 4.2. Welding Performance

Weldability of aluminum alloy is increased by using the laser cleaning to remove the oxide films from the aluminum alloy surfaces. The researches found that the laser cleaning can remove most lubricants and contaminants attached to the surface, eliminate hydrogen and other gases generated by coatings, lubricants, and surface contaminants, and reduce the source of hydrogen generated by micropores during welding, so as to significantly improve the welding quality [8,62,70,71]. By comparing the weld seam porosity on an aluminum alloy surface after three pretreatments, Zhou et al. [62] found that after laser cleaning in air, the weld seam porosity of the aluminum alloy was reduced to 2.91% from the maximum value of 9.68% when the surface was untreated. After the laser cleaning in argon, the weld seam porosity was further reduced to just 1.59%. Haboudou et al. [70] also obtained similar results, as shown in Figure 8. With all surface preparations, the porosity rate tends to be lowered at both welding speeds, but laser cleaning has the most noticeable impact, practically completely suppressing pores in A356 (the residual porosity rate is less than 2%) and decreasing them by a factor of two on AA5083. Furthermore, Chen [72] found that after the laser cleaning, the oxide film on aluminum alloy surfaces formed a micro-nanostructure, and the welding depth initially decreased and then increased to a gradual level during welding, which was 23.4% higher than that of the uncleaned surface and 10.1% higher than that of the mechanically polished surface.

### 4.3. Adhesive Performance

The paint layers are removed from aluminum alloy surfaces for maintenance and respraying, and the removal effect of the paint layer is the main factor to be considered. The adhesion performance of the resprayed paint layer is also an issue following laser cleaning. The study showed that the adhesion of the substrate coating is significantly improved after the laser cleaning, and it is superior to the mechanical grinding [68]. The primary reason is that the substrate surface roughness decreases after the laser cleaning, and, consequently, so does the coating shear force under the same load. Furthermore, the laser ablation treatment produces a new oxide layer with a higher surface activity on the aluminum alloy substrate [71,73], promoting the complete wetting of the adhesive and enhancing the bond strength.

### 4.4. Other Performances

The laser cleaning improves the corrosion resistance, welding performance, adhesion performance, tensile and bending properties, surface microhardness, and friction and wear performance of the aluminum alloy [35,73,74,75,76]. Wang et al. [35] found that the surface microhardness of the 7075 aluminum alloy increased by 5.62–8.45% as a result of the shockwaves generated by plasma explosion and air and water explosion in the strip defect, as shown in Figure 9. Yuan et al. [76] used a pulsed laser to remove the oxide layer from the surface of an AA2024 aluminum alloy used for the aerospace industry. The results showed that the laser cleaning could improve the friction and wear performance of the AA2024 surface by altering its microstructure. Furthermore, the laser cleaning produced a hardened layer on the AA2024 surface and increased its microhardness.

## 5. Real-Time Detection of Laser Cleaning

The laser cleaning effect of oxide films or paint layers on aluminum alloy surfaces relates to the laser space, beam parameters, and the physical properties of the material being cleaned. To ensure that the cleaning effect can meet the requirements, it is necessary to conduct the real-time detection of the laser cleaning process. Simultaneously, the real-time detection of laser cleaning is critical for realizing laser-controlled and intelligent cleaning. The real-time detection technology of laser cleaning focuses on the acoustic signals, laser-induced breakdown spectroscopy (LIBS), and robot vision [77,78,79,80,81,82,83,84,85,86,87].

A laser irradiates the oxide film or paint layer on the aluminum alloy surface. The oxide film or paint layer on the aluminum alloy surface rapidly vaporizes and volatilizes when the energy received by the surface is greater than the material vaporization point. The vaporized particles keep absorbing energy. When the surface absorbs the energy above the ionization energy, the ionization will occur, resulting in the formation of a plasma cloud with a high temperature and high-pressure that will produce the shock waves [77]. When the surface absorbs the energy below the material melting point, the material will be expanded due to the increase of the temperature. In this case, the periodic deformation of the surface layer caused by the nature laser pulses will radiate the regular thermoelastic acoustic waves [78]. Whether a plasma acoustic wave or a thermoelastic acoustic wave, it is related to the material thermal properties, optical properties, and incident laser properties. During the laser cleaning process, the thermal and optical properties of the material change, resulting in the changes in the acoustic signal. As a result, it is possible to monitor the online cleaning process by detecting the acoustic signal generated by laser cleaning. Specifically, a microphone can be used to collect the acoustic signal from the cleaning surface when the oxide film or paint layer is cleaned. An analog signal can be collected and converted into a digital signal using an audio card and then be sent to a computer for signal analysis. Studies have shown that the amplitude of the acoustic signal varies with the paint thickness [79]. When the laser interacts with the paint layer, the amplitude of the acoustic signal is high and steady. However, after the paint layer is removed, the laser interacts with the aluminum alloy substrate with the high reflectivity and low absorption, which causes the amplitude of the acoustic signal to drop. Therefore, the change in amplitude of the acoustic signal throughout the cleaning process may be used to determine indirectly if the paint layer has been removed from an aluminum alloy surface. Furthermore, the frequency of the acoustic signal can be used to determine the cleaning threshold when the laser cleans the paint layer on the aluminum alloy surface.

Zou [80] found that when the laser energy is below the cleaning threshold, the maximum amplitude frequency of the received acoustic signal is all above 0.04 kHz. Once the laser energy density is equal to or above the cleaning threshold, the maximum amplitude frequency is always 0.04 kHz. Figure 10 illustrates this phenomenon graphically. We define 0.04 kHz as the characteristic frequency of acoustic signal monitoring in the laser cleaning process. Specifically, the characteristic frequency is relevant to the type of the substrate. Therefore, we can propose an acoustic monitoring technology to assess whether the paint has been removed by analyzing the acoustic signal. Compared with other monitoring technology, acoustic monitoring has the advantages of a broad application range, high efficiency, and the cheap running cost. However, it is noted that the audible frequencies could also carry lots of noises, which might reduce their accuracy in the commercial setting.

Yang et al. [81] adopted the LIBS technology to conduct spectral analysis of the characteristic elements in the paint layers of an aircraft aluminum alloy skin multicoating structures. They established the internal correlation between the quantity and thickness of paint layer removal and the change in the LIBS spectrum based on the signal interpretation. It can be judged in real time as to whether the topcoat and primer have been removed based on the presence of the characteristic spectral peaks of FeI and TiI. Furthermore, the thickness of paint layer removal may be obtained by monitoring the spectral signal intensity of CaI. Similarly, the LIBS can detect the effect of the laser cleaning of the oxide film from aluminum alloy surfaces online [82]. Collimating optics were used to collect the plasma signals generated in the laser cleaning process, which were then sent to the spectrometer via the optical fiber for analysis and processing. The cleaning effect was judged according to the change in the intensity of the O–Al characteristic spectral line. However, a plasma generation is necessary for the LIBS to detect the laser cleaning effect of the oxide films on aluminum alloy surfaces online.

Machine vision technology also can realize the real-time detection during the laser cleaning of oxide films on aluminum alloy surfaces. Shi et al. [83] designed a real-time monitor system for laser cleaning aluminum alloy based on machine vision, used the Visual Studio 2015 as the development platform, coupled various machine vision detection algorithms, and realized the accurate segmentation and rapid positioning of qualified and unqualified areas. The detection accuracy can reach 0.03 mm, and the total processing time of a single image is 400 ms. Specifically, Shi [83] used a CCD camera and an auxiliary light source to synchronously collect the laser-cleaned image of the aluminum alloy surface and then imported the image into the machine vision-based real-time monitor system for laser cleaning aluminum alloy for image processing and calculation. The image is filtered using the median filter method first. To reconstruct a more realistic original image, the image is de-illuminated using the Retinex single-scale algorithm [84] based on the color constancy perception theory. The image is then subjected to Gamma transformation to correct the grayscale of the image and enhance the contrast after being de-illuminated by the Retinex algorithm in order to make it easier to distinguish the light and dark features of the image. Next, the maximum inter-class variance algorithm Otus [85] is used to dynamically segment the light and dark regions in the image. After segmentation, the dilation algorithm is used to remove the over-segmented regions, and the sections with too little an area are also removed to obtain the segmented image. The dark area is the nonconforming region. The Canny edge identification technology [86] and the connection threshold search algorithm [87] are used to identify the nonconforming region size and precise location.

## 6. Summary and Outlook

This paper reviews the mechanism of laser cleaning, the influence of different laser process parameters on the cleaning effect, optimization of the substrate performance, and the real-time detection technology of laser cleaning.

The ablation gasification, vibration stripping, and explosion stripping are the three mechanisms of laser cleaning. The oxide film or paint layer on the surface of the aluminum alloy is directly vaporized and removed when the laser energy received by the surface of the aluminum alloy is higher than the vaporization point of the material. Otherwise, the oxide film or paint layer is peeled off from the substrate under the influences of the expansion and shock waves.

The effect of removing the oxide film or paint layer from the aluminum alloy surface depends critically on laser process parameters. Increasing the laser power, energy density, repetition rate, and pulse width or slowing down its cleaning speeds, scanning speeds and the focused spot area will promote the effect of cleaning. However, the substrate will be ablated when the laser process parameters continue to increase or reduce. Therefore, it is essential to determine the laser cleaning and damage thresholds before laser cleaning. The corrosion resistance, welding performance, adhesion performance, tensile and bending properties, surface microhardness, and friction and wear performance were enhanced after laser cleaning using suitable laser process settings.

Real-time detection technologies, such as acoustic signals, laser-induced breakdown spectroscopy (LIBS), and robot vision, would be a wise choice for implementing laser-controlled and intelligent cleaning.

Although the laser can successfully remove the oxide film and paint layer on the surface of aluminum alloy, further study on the following topics can be done in the future to encourage the use of laser cleaning technology.

(1) Perfect the laser cleaning mechanism. Although there are various mechanisms under different laser cleaning conditions according to the existing literature, research on how different mechanisms affect the cleaning effect has not been taken into account. There is a risk of secondary oxidation or ablation of the substrate during laser cleaning. Thus, follow-up research on the laser cleaning mechanism should concentrate on the cross-coupling of optics, heat, and mechanics between the laser beam and the materials. Furthermore, to realize the controlled selection of the mechanism in the cleaning process, it is also necessary to investigate the quantitative link between the laser cleaning mechanism, the materials, and the laser parameters.

(2) Establish laser cleaning process and evaluation standards. The composition, thickness, density, and adherence of oxide film or paint layer on the aluminum alloy surface vary in various applications. When utilizing laser cleaning, the optimal process parameters change. In addition, because each study team used a different assessment method to determine the efficient removal of the oxide film and paint layers, the optimal cleaning process parameters were not consistent. In order to achieve the best possible configuration of laser cleaning parameters, it is imperative to intensify quantitative research on the physical and chemical characteristics of the oxide film and paint layers, conduct numerous coupled studies on the process parameters of laser cleaning, and standardize cleaning evaluation criteria.

(3) Improve the applicability of online detection of laser cleaning. Currently, the online detection of laser cleaning focuses on acoustic signals, spectral signal detection, and robot vision. Acoustic signals are widely used and affordable, yet ambient noise has a considerable impact on them. The detection efficiency of a spectral signal is high, but it requires plasma generation, which is inappropriate for low energy density. Robot vision is perfect for image processing, but there is still room for improvement in processing speed. In order to further improve the applicability of the real-time detection of laser cleaning, on the one hand, the advantages of acoustic monitoring technology, spectral signal detection technology, and machine vision could be supplemented. On the other hand, more advanced detection technology should be explored.

(4) Increase the research on intelligent laser cleaning technology. Do research on laser intelligent cleaning-related feedback algorithms. Based on the results of this research, we can realize the high precision orientation of laser cleaning heads and dynamic adjustment of cleaning parameters by using advanced sensing and online detection technology for the laser cleaning, together with robotic arms. It can effectively advance the industrial application of the laser cleaning from the current two-dimensional cleaning to the complex three-dimensional surface cleaning.

## Figures and Tables

**Figure 1 materials-15-05469-f001:**
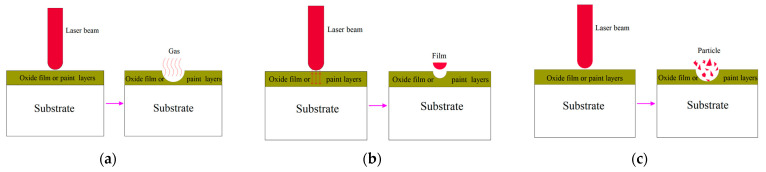
The mechanism of laser cleaning. (**a**) ablation gasification; (**b**) vibration stripping; (**c**) explosion stripping.

**Figure 2 materials-15-05469-f002:**
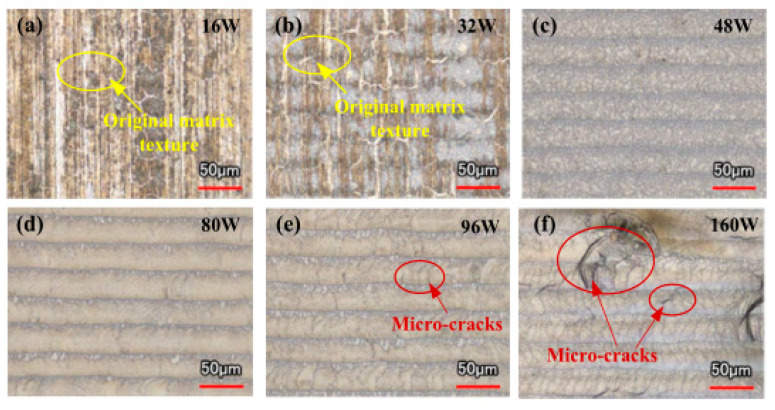
Morphology of laser-cleaned 7075 aluminum alloy with different laser powers [50]. (**a**) 16 W; (**b**) 32 W; (**c**) 48 W; (**d**) 80 W; (**e**) 96 W; (**f**) 160 W.

**Figure 3 materials-15-05469-f003:**
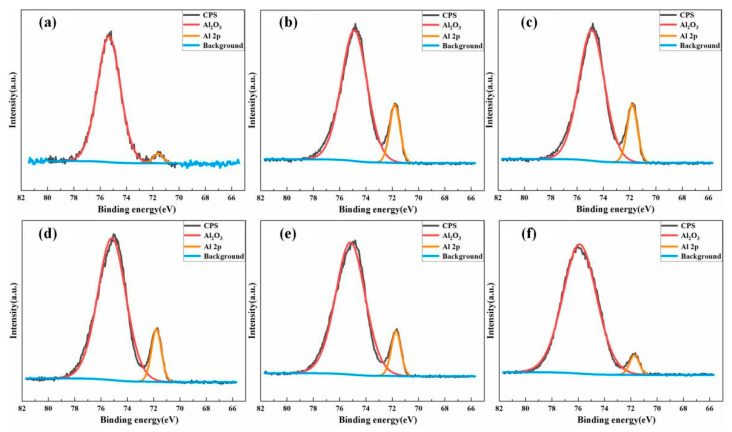
Detailed Al 2p X-ray photoelectron spectroscopy. (**a**) untreated and laser-cleaned with (**b**) 3.5 J/cm^2^; (**c**) 17.5 J/cm^2^; (**d**) 26.25 J/cm^2^; (**e**) 35 J/cm^2^; (**f**) 52.5 J/cm^2^ [33].

**Figure 4 materials-15-05469-f004:**
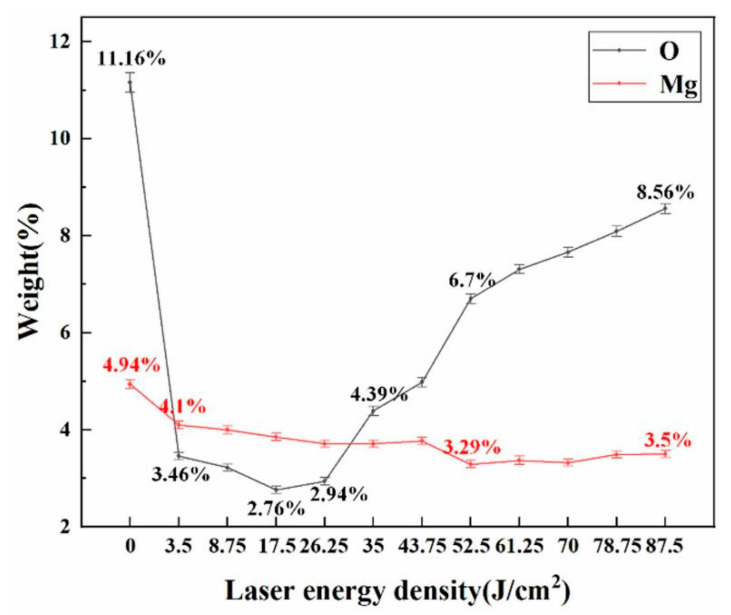
Changes in surface elements with different laser energy densities [33].

**Figure 5 materials-15-05469-f005:**
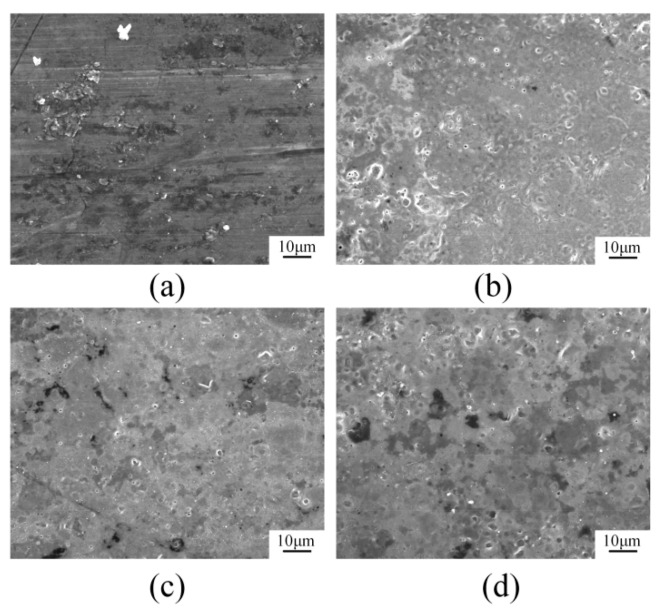
Scanning electron microscopy surface microstructure of aluminum alloy cleaned using different cleaning speeds. (**a**) reference sample; (**b**) 624 mm·min^−1^; (**c**) 246 mm·min^−1^; (**d**) 60 mm·min^−1^ [51].

**Figure 6 materials-15-05469-f006:**
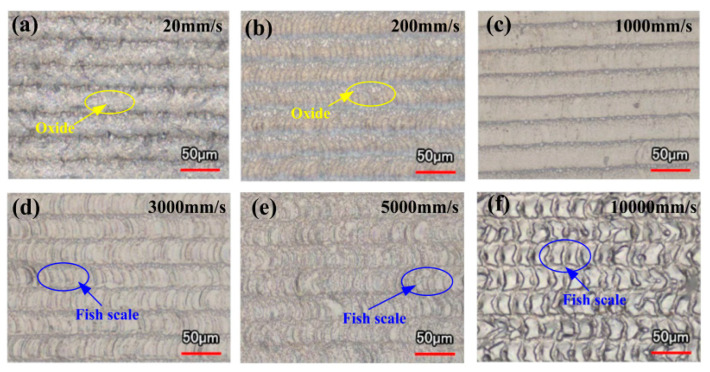
Morphology of a laser-cleaned surface with different scanning velocities [50]. (**a**) 20 mm/s; (**b**) 200 mm/s; (**c**) 1000 mm/s; (**d**) 3000 mm/s; (**e**) 5000 mm/s; (**f**) 10,000 mm/s.

**Figure 7 materials-15-05469-f007:**
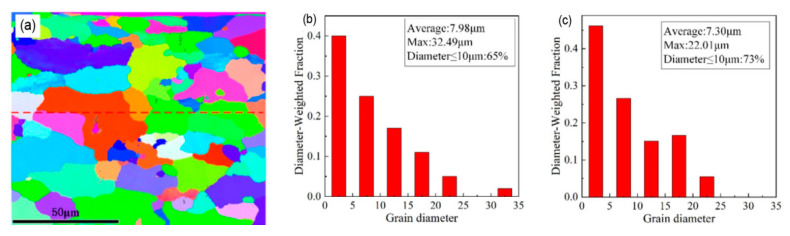
Statistics of grain size of aluminum alloy substrate after laser cleaning. (**a**) electron backscatter diffraction results; (**b**) entire region; (**c**) upper region [68].

**Figure 8 materials-15-05469-f008:**
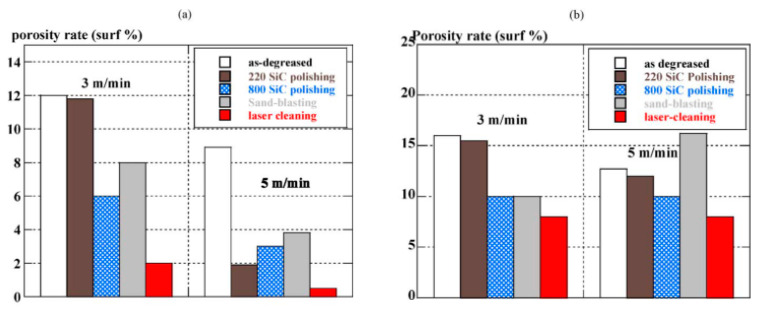
Influence of surface preparation on the porosity rate with two different welding speeds at 4 kW in (**a**) A356 and (**b**) AA5083 [70].

**Figure 9 materials-15-05469-f009:**
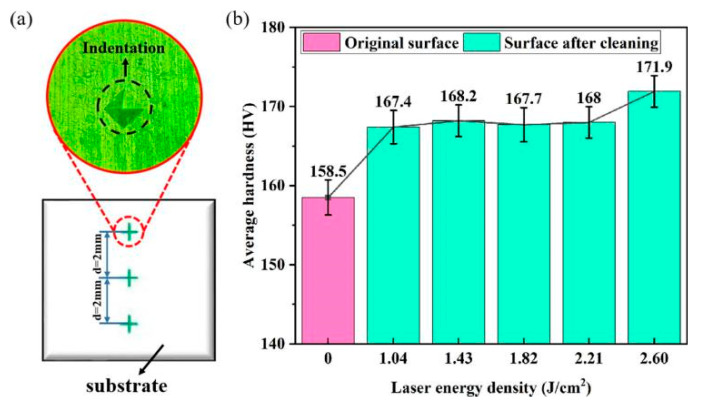
(**a**) Microhardness indentation and test point position; (**b**) change in microhardness values before and after laser cleaning [35].

**Figure 10 materials-15-05469-f010:**
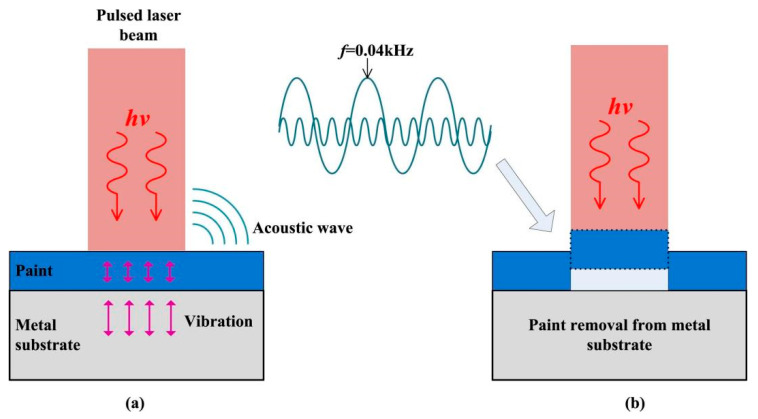
Schematic of the explanation for characteristic frequency. (**a**) The generation of the acoustic wave and (**b**) paint removal from the metal substrate [80].

**Table 1 materials-15-05469-t001:** Optimization of laser beam parameters.

Substrate	Laser Type	Material Removed	Thickness	Parameter Type	Range	Removal Threshold	Evaluation Method	Uncleaned	Cleaned	Ref.
7050	Nanosecond pulsed fiber laser	Oxide film	/	Power (W)	16–160	80	Oxygen content	8.18%	1.36%	[50]
5A12	Nd:YAG	Oxide film	/	Power (W)	50–110	98	Oxygen content	14.4%	8.3%	[51]
6061	Nanosecond pulsed fiber laser	Oxide film	/	Power (W)	15–25	20	Oxygen content	44.86%	7.72%	[52]
5083	Nanosecond pulsed fiber laser	Oxide film	5 μm	Power (W)	26–74	42	Oxygen content	50.85%	4.79%	[53]
2024	Nanosecond pulsed fiber laser	Paint layer	50 μm	Power (W)	10.5–25.5	16.5	Surface roughness	/	1.615 μm	[54]
2024	Nanosecond pulsed fiber laser	Paint layer	150 μm	Power (W)	400–600	556.17	Substrate damage	0	0	[12]
AA5083	Nanosecond pulsed fiber laser	Oxide film	/	Fluence (J/cm^2^)	3.5–8.5	7.1	UV–vis spectra	~25%	~75%	[55]
5083	Nanosecond pulsed fiber laser	Oxide film	/	Fluence (J/cm^2^)	3.5–35	17.5	Oxygen content	11.16%	2.76%	[33]
AA7024– T4	Nanosecond pulsed fiber laser	Oxide film	/	Fluence (J/cm^2^)	3.5–11.3	7.1	MgAl_2_O_4_ content	11.05%	0	[56]
2024	Nd:YAG	Paint layer	130 μm	Fluence (J/cm^2^)	2–6	5	Al content	0	79.5%	[13]
7075	Nanosecond pulsed fiber laser	Paint layer	40 μm	Fluence (J/cm^2^)	8–28	24	Temperature	0	769 K	[57]
Al–Mg series	Nanosecond pulsed fiber laser	Paint layer	40 ± 5 μm	Fluence (J/cm^2^)	1.2–8	3.2	Oxygen content	1.64%	1.57%	[58]
2024	Nanosecond pulsed fiber laser	Paint layer	100 μm	Fluence (J/cm^2^)	5–40	25	Temperature	0	509 °C	[59]
2024	Nanosecond pulsed fiber laser	Paint layer	50 μm	Fluence (J/cm^2^)	14.15–35.39	21.23	Oxygen content	/	0.96%	[60]
6160	Nanosecond pulsed fiber laser	Oxide film	/	Frequency (kHz)	40–140	100	Oxygen content	/	4%	[61]
2024	Nanosecond pulsed fiber laser	Paint layer	50 μm	Frequency (kHz)	20–45	30	Removal rate	0	100%	[54]
5083	Nanosecond pulsed fiber laser	Oxide film	5 μm	Pulse width (ns)	26–240	100	Oxygen content	/	5.03%	[53]

**Table 2 materials-15-05469-t002:** Optimization of laser space parameters.

Substrate	Laser Type	Material Removed	Thickness	Parameter Type	Range	Removal Threshold	Evaluation Method	Uncleaned	Cleaned	Ref.
5A12	Nd:YAG	Oxide film	/	Cleaning speed (mm min^−^^1^)	60–1242	246	Oxygen content	14.4%	8.3%	[51]
2219	Nanosecond pulsed fiber laser	Oxide film	24 μm	Cleaning speed (mm min^−^^1^)	500–3500	3500	Oxygen content	21.67%	2.8%	[63]
7075	Nanosecond pulsed fiber laser	Oxide film	/	scanning speed (mm min^−1^)	20–3000	1000	Surface roughness	/	0.122 μm	[50]
5083	Nanosecond pulsed fiber laser	Oxide film	5 μm	scanning speed (mm min^−^^1^)	3000–7000	4725	Oxygen content	50.85%	3.89%	[53]
2024	Nanosecond pulsed fiber laser	Paint layer	50 μm	scanning speed (mm min^−^^1^)	200–1200	600	Surface roughness	/	1.205 μm	[54]
7075	Nanosecond pulsed fiber laser	Paint layer	40 μm	scanning speed (mm min^−1^)	1600–2400	2000	Surface compound	Appeared	Disappeared	[57]
2024	Nanosecond pulsed fiber laser	Paint layer	200 μm	Defocus (mm)	−4–+4	0	Surface topography	/	Gasify	[64]
6005A	Nanosecond pulsed fiber laser	Paint layer	/	Defocus (mm)	−3–+3	0	Surface color	Black	Bright white	[65]

## Data Availability

Not applicable.

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
