# Peer review of "Research Progress and Challenges in Laser-Controlled Cleaning of Aluminum Alloy Surfaces"

_materials, 2022, doi:10.3390/ma15165469_

Round 1
Reviewer 1 Report
Reviewer comments
The paper titled,” Research progress and challenges in laser controlled cleaning of aluminium alloy substrates”, discusses about the literature review based on laser cleaning of aluminium alloy. I studied this paper carefully. Following are my comments:
1. Abstract should be revised and it is suggested to remove the word “we”.
2. This paper have lots of grammatical mistakes. These should be corrected.
3. Authors have written at many places like surface’s roughness, beam’s power, substrates’s surface………etc. This way of writing should be avoided.
4. Aluminium is highly reflecting material. Justify the application of laser cleaning of aluminium alloy.
5. Mention the optimum power range and laser fluence used in laser cleaning of aluminium alloy.
6. Section 4.3 and section 4.4 are repeated, “Adhesive performance”.
7. Conclusion parts need revision. It should be more precise.
Overall this paper is not suitable for publication in present form.

Reviewer 2 Report
Please find my comments in the attached pdf file.

Reviewer 3 Report
The manuscript: “Research progress and challenges in laser–controlled cleaning of aluminum alloy surfaces” summarizes some research about laser cleaning of aluminium alloy surfaces. The approach is interesting with a relevant discussion of process parameters and challenges. I recommend a major revision since there are some weaknesses in explanations that obscure the understanding and make the conclusion and outlook less convincing.
· It is problematic that the conclusions regarding the individual parameters influence on the cleaning process are based on specific but different processing conditions presented from different labs and research teams. The conclusions should be followed by a problematizing discussion that can convince the readers that the revealed trends are generic, not depending on the contamination at hand or the base alloy or something similar, if this now is true.
· The manuscript describes the following process parameters: laser power (total power), laser energy density (radiant exposure), laser repetition frequency (pulse frequency), laser cleaning speed (I assume travel speed), laser scanning speed (this is problematic! Is it a local scanning pattern in a global reference frame that travels with the cleaning speed? If so, it is more appropriate to define a velocity and scanning pattern, I assume it is a repetitive pattern), pulse width (temporal), and laser defocus. Since these parameters are coupled there is a need to also define the following: the laser power density distribution, beam Rayleigh length. A discussion about defocus and radiant exposure is not meaningful without this information for example. Other parameters that are relevant but not explained or defined are the laser interaction time (related to pulse duration/scanning- and travel speed) and if there occur multiple exposures on the same surface area during processing. I also think that laser wavelength is a relevant parameter but not discussed.
· It is obvious that the different parameters discussed are coupled. I lack a discussion about this issue since there are suggestions in the summary and outlook section that talks about optimization and feedback control. If there is no information about this coupling that can be complemented to the manuscript at least there should be a discussion about how to deal with this.
I have found some details in the text that needs attention.
Page 1, raw 29 Abstract: “This paper also identifies the real–time detection methods in the laser cleaning process and investigate the research and development direction of the laser cleaning technology.” This sentence suggests that all possible real-time detection methods relevant for laser cleaning have been identified in this paper. This is likely not true. For example, in the outlook the authors suggest the use of infrared imaging. I suggest that this sentence should be rewritten.
Page 2, Figure 1. The pictures in the centre of the illustration are too tiny to be recognized. Either remove or redo it.
Page 3, raw 97 and 98: “controllable direction, and strong convergence ability” – replace with spatial coherence
Page 3, raw 114: regarding evaporation (Evaporation is vaporization that happens at temperatures below the boiling point. Should be insignificant in laser cleaning) use the word vaporization only.
Page 12, raw 404-405: Give reference to what is meant by “welding grade III”
Page 13, row 453-455 “As the coating on the aluminum alloy surface has a higher absorption coefficient for laser energy than the substrate, a microphone can be used to collect the acoustic signal from the cleaning surface when the paint layer is cleaned”- this claim is not logic. The acoustic emission is due to pressure waves and has nothing to do with different absorptivity in different materials. I suggest that this sentence should be rewritten.
Pages 13 and 14, raws 446-477: is the acoustic emission airborne or waves in solid? This has very practical implications. Explain more in details.
Pages 14, raws 465-476: The reasoning about acoustic signals is not logic. The value 40 Hz that is mentioned in this context, is it a universal constant or is it application dependent? “Furthermore, time–amplitude, frequency–amplitude (FA), and time–frequency (TF) spectroscopies have different properties at varying laser fluences.” – this sentence is problematic since at first, the time-amplitude representation is not a spectrum. Secondly what is meant by spectroscopies?
Page 14, raw 492: replace “An optical signal collector” with for example “A collimating optics”
Page 15, raw 498-507: This paragraph is very problematic. Several algorithms are listed with no references. How can you segment an image accurately by combining Otsus method with morphology? What is the relevant information and what is quickly location? “Li et al. proposed an intelligent laser cleaning method based on machine vision, and after a series of tests, the accuracy of particle swarm optimization–reasoning or a support vector machine (PSO–SVM) reached 92.5% [82].” – I can not understand the message in this sentence. Please explain.
Page 16, raw 552-553: Here suggests some temperature monitoring. In general, it is a very complex task to measure temperatures by infrared thermography. Especially when you have metallic surfaces at different oxidation states. My question is why temperature is the most relevant quantity. I believe that thermograms can be used to get information related to heat radiation that can be linked to process deviations and relevant quality issues. I suggest that this sentence should be rewritten to better explain what the authors really mean.
Page 16, raw 557-559: “For the online inspection of laser cleaning on complex parts and large components, the research and application of robot vision technology will be the focus of future laser cleaning.”- How have the authors reached this conclusion? There is no motivation to that in the manuscript.
Page 16, raw 569: I suggest replacing “algorithm feedback” with “feedback algorithm”
Round 2
Reviewer 1 Report
Reviewer’s Comments
The present paper titled, “Research progress and challenges in laser controlled cleaning of aluminium alloy surfaces”, discusses about the application of laser energy for removal of oxide layers and paints from the aluminium alloys surface before further welding or other processing. This topic is interesting, but authors have presented this paper in poor manner. Therefore, I cannot recommend for publication of this paper in the journal. Followings are my comments:
1. This paper does not reveal new knowledge for future researchers working in the area of laser cleaning. This is just describing the collection of literature review without rigorous study.
2. Abstract is presented in poor way.
3. English language of this paper is not clear.
4. It indicates superficial study of literature based on laser application for surface cleaning/ paint removal.
5. Summary and outlook is written poor way. It should be more precise.
6. The summary and outlook should contain the future scope and challenges of laser cleaning process and their remedial measures.
7. Already laser processing machines are automatic controlled systems with CNC controller for movement of worktable and torch. Then point on (3) in the summary is not revealing any new knowledge.
Overall, this paper should be rejected.

Round 3
Reviewer 1 Report
Reviewer comments
The paper titled,” Research progress and challenges in laser controlled cleaning of aluminium alloy substrates”, discusses about the literature review based on laser cleaning of aluminium alloy. I studied this paper carefully. Following are my comments:
1. This paper have lots of grammatical mistakes. These should be corrected.
2. Authors have written like explosion’s expansion, substrates’s surface………etc. This way of writing should be avoided.
3. Summary and outlook parts need revision. It should be more precise.
Overall this paper is not suitable for publication in present form.
